# Assessment of the Genetic Diversity of Rice Germplasms Characterized by Black-Purple and Red Pericarp Color Using Simple Sequence Repeat Markers

**DOI:** 10.3390/plants8110471

**Published:** 2019-11-04

**Authors:** Jae-Ryoung Park, Won-Tae Yang, Yong-Sham Kwon, Hyeon-Nam Kim, Kyung-Min Kim, Doh-Hoon Kim

**Affiliations:** 1Division of Plant Biosciences, School of Applied Biosciences, College of Agriculture and Life Science, Kyungpook National University, Daegu 41566, Korea; icd92@naver.com; 2Department of Molecular Genetic Engineering, College of Natural Resources and Life Science, Dong-A University, Busan 49315, Korea; wtyang@dau.ac.kr (W.-T.Y.); yskwon@dau.ac.kr (Y.-S.K.); wiseman311@naver.com (H.-N.K.)

**Keywords:** rice (*Oryza sativa* L.), simple sequence repeat (SSR), cultivar identification, molecular marker, plant variety protection

## Abstract

The assessment of the genetic diversity within germplasm collections can be accomplished using simple sequence repeat (SSR) markers and association mapping techniques. The present study was conducted to evaluate the genetic diversity of a colored rice germplasm collection containing 376 black-purple rice samples and 172 red pericarp samples, conserved by Dong-A University. There were 600 pairs of SSR primers screened against 11 rice varieties. Sixteen informative primer pairs were selected, having high polymorphism information content (PIC) values, which were then used to assess the genetic diversity within the collection. A total of 409 polymorphic amplified fragments were obtained using the 16 SSR markers. The number of alleles per locus ranged from 11 to 47, with an average of 25.6. The average PIC value was 0.913, ranging from 0.855 to 0.964. Four hundred and nine SSR loci were used to calculate Jaccard’s distance coefficients, using the unweighted pair-group method with arithmetic mean cluster analysis. These accessions were separated into several distinctive groups corresponding to their morphology. The results provided valuable information for the colored rice breeding program and showed the importance of protecting germplasm resources and the molecular markers that can be derived from them.

## 1. Introduction

Rice (*Oryza sativa* L.) is one of the most widely cultivated crops in the world; distributed across a variety of climates and regions, it has developed a diverse array of genotypes and phenotypes. Koreans have historically used rice as a major food source. However, in recent years, both the production and consumption of rice have been in decline. In 2017, rice production was 3.972 million tons, down 5.3% from 4.197 million tons the previous year, and the annual rice consumption per capita in 2017 was 70.9%, down 0.4% from the previous year. The continuous decrease in the production and consumption of rice in Korea means that the industry is not stable [1]. One reason for the annual decline in rice consumption is the increase in the number of people seeking meat and salad alternatives because of a preference for a westernized diet [2]. Therefore, to increase the consumption of rice in Korea, processed food products using a rice base have been developed. Colored rice varieties such as black-purple rice are particularly popular for this purpose because of their unique color and aroma [3,4]. To increase rice consumption and production, high-quality rice containing the functional materials preferred by producers and consumers are also cultivated [5]. Rice varieties are largely classified according to the main areas of their cultivation, such as *Oryza sativa* L. ssp. *japonica, Oryza sativa* L. ssp. *indica*, and *Oryza sativa* L. ssp. *javanica* [6,7,8]; these can be utilized to further breed new, high-quality varieties. Despite the abundance of rice genetic resources globally, the genetic similarity of rice varieties has increased, and the variety of species has continued to decrease, since only a few specific varieties are utilized for breeding [9,10]. In the process of producing new varieties, genetic diversity was greatly reduced in both wild and cultivated rice. At present, various rice breeding institutes in Korea and abroad are trying to breed high-quality rice varieties; among the cultivars developed with the artificial hybridization method rather than the transgenic technique are “Tongilbyeo” and “Naepungbyeo” [11,12]. These are usually made by crossing several different varieties of ssp. *japonica*. It is assumed that the currently developed varieties are genetically very similar. As the minimum genetic distance between the varieties is very small because of the genetic resources used for cultivation, the varieties of rice in Korea are not very diverse, and seed disputes related to genotype use and the protection of breeders’ rights frequently occur. To overcome these problems, we have constructed a database for each species using molecular markers and established a genetic linkage to known varieties [13]. The International Union for the Protection of New Varieties of Plants (UPOV) also provides guidelines for database construction of varieties using molecular markers [14]. Assessing the diversity of rice genetic resources involves identifying phenotypes, analyzing biochemistry, and evaluating DNA diversity [15,16]. The evaluation methods involving the identification of phenotypes and biochemical characteristics are not necessarily reliable, as they are environmentally influenced, labor intensive, and costly [17,18]. However, evaluating genetic diversity based on DNA is the most widely used evaluation method in terms of repeatability, stability, and reliability. Several techniques have been developed to analyze the genetic diversity of rice varieties based on DNA, such as the analysis of restriction fragment length polymorphism (RFLP) using restriction enzymes, random amplified polymorphic DNA (RAPD), amplified fragment length polymorphism (AFLP), simple sequence repeats (SSR), and single nucleotide polymorphisms (SNP) utilizing the polymerase chain reaction (PCR) [19,20]. The RAPD and AFLP techniques are not easily reproducible and the markers are dominant, so there are many limitations in analyzing the relationship of genetic resources. However, since SSR markers exhibit high polymorphism and show co-dominant status, they can be used to identify heterozygosity. SSR markers, also called DNA microsatellites, are regions of DNA (often forming part of the non-coding regions) where sequences of one to five nucleotides are repeated, and they are uniformly distributed in the genomes of most eukaryotes. The SSR sequences found in plants are frequently made up of AT and GA nucleotide repeats. The relationships of genetic resources have been analyzed using SSR markers in other crops such as melon (*Cucumis melo* L.), watermelon (*Citrullus lanatus*), and corn (*Zea mays*) [21,22,23]. SSR markers representing the entire genome have been developed and commercialized for rice with an average distance of 3–4 cM between markers [17,24,25], but few have been used to identify rice varieties to protect the breeders’ rights. Colored rice has many antioxidants. At present, the development of colored rice with high antioxidant content is being made continuously. Therefore, in this study, if we constructed a DNA database of colored rice gene source by using SSR markers, this study was expected to be helpful for the improvement of colored rice varieties in the future. Also, in this study, we developed a core marker set for the identification of rice varieties using SSR markers and constructed a DNA profile database for rice varieties using the developed markers. To characterize the SSR markers with a high degree of polymorphism in the rice genetic resources, allele characteristics of 600 SSR markers reported in the rice genome database (http://www.gramene.org/) were used. The best markers were selected and analyzed for gene affinity with 548 colored rice genetic resources (376 black-purple rice samples, 172 red pericarp rice samples) collected both at home and abroad.

## 2. Results

### 2.1. Microsatellite Analysis

Assessing the genetic diversity of our resources is an essential process for efficiently preserving biodiversity and for characterizing and exploiting them. We used 600 microsatellite primers, reported in the rice genome database (http://www.gramene.org/), to select microsatellite markers suitable for rice germplasm identification and genetic diversity assessment. The microsatellite markers were amplified in 11 different varieties: “Hopum,” “Hwayeong,” “Dami,” “Sindongjin,” “Odae,” “Chucheong,” “Saechucheong,” “Junam,” “Ilpum,” “Ilmi,” and “Gopum.” Polymorphisms of primers and characteristics of the alleles were examined after removing those that showed the dominant form, had no obvious bands, or had low repeatability when the PCR was performed; 16 microsatellite markers were deemed suitable (KRF24, RM8085, RM72, KRF16, HvSSR02-86, RM18821, RM12834, RM20754, RM26063, RM333, RM15641, RM26730, RM19731, KRF15, KRF17, RM24946). The 16 microsatellite markers showed co-dominance, high repeatability, and high polymorphism rates and were selected to evaluate the diversity of the rice germplasm resources (Figure 1 and Table 1). To investigate the genetic polymorphisms of the chromosomal genetic resources, the fluorescent labels FAM, VIC, NED, and PET were attached to the forward 5′-ends of the 16 microsatellite markers selected above. PCR and electrophoresis were carried out with the rice germplasm DNA using an automatic nucleotide sequencer, and the degree of polymorphism was examined. A total of 409 alleles were amplified with the selected microsatellite markers. The number of alleles found in each varietyranged from 11 to 47, with an average of 25.6 alleles per variety. The polymorphism information content (PIC) values ranged from 0.855 to 0.964, and the average was very high at 0.913. High PIC values indicate that the selected microsatellite markers are efficient at evaluating a large number of genetic resources. In this study, the mean PIC value of the microsatellite markers used to assess the diversity of rice genetic resources was 0.913, which was much higher than the PIC values of microsatellite markers used in other studies [26,27,28]. These results indicate that the microsatellite markers selected for use in this study are very suitable and efficient for assessing the genetic diversity of many different rice genetic resources. Several studies have been conducted to evaluate genetic diversity using microsatellite markers in rice [17,24,25]. However, because of the abundance of rice genetic resources, and as only a fraction of these resources have been analyzed [29], it is necessary to conduct more such studies. In addition, since the PIC values of the microsatellite markers used in this study were much higher than those in other studies [30], it appears that the microsatellite markers used for the analysis of the 548 rice genetic resources in this study could be used more effectively to establish the genetic relationship of rice genetic resources and to create a database for each species.

### 2.2. Hierarchical Cluster Analysis

The unweighted pair-group method with arithmetical average (UPGMA) dendrogram was constructed using the MEGA7 program and Jaccard’s distance coefficients using the results of the PCR with the 16 microsatellite markers selected from the 548 colored rice genetic resources (Appendix A). The analysis classified the 548 rice genetic resources into four major groups (I, II, III, and IV) with seven subgroups (I-1, I-2, I-3, II, III, IV-1, and IV-2). In addition, as the black-purple rice and red pericarp rice were not clearly distinguished from each other, they were mixed and classified according to the environmental conditions of the areas where each genetic resource was grown (Figure 2). The 548 rice genetic resources used in this study included 337 varieties in Group I, 13 varieties in Group II, 144 varieties in Group III, and 54 varieties in Group IV. Group I included 18 varieties from Malaysia, five from Myanmar, two from Vietnam, two from Bhutan, five from Sri Lanka, 95 from Indonesia, 43 from Japan, 78 from China, three from Cambodia, three from Taiwan, 26 from Thailand, two from Pakistan, 23 from the Philippines, and 32 from Korea. Group I included most of Asia and could be divided into three subgroups (I-1, I-2, I-3). Subgroup I-1 included mostly South Asian countries such as Malaysia, Indonesia, and Cambodia. Subgroup I-2 included East Asian countries such as Japan and Korea, and Subgroup I-3 included Malaysia, Bhutan, Sri Lanka, Indonesia, Japan, China, Thailand, the Philippines, and South Korea. Group II contained one variety from Indonesia, 11 from Japan, and one from Korea. Group III included one variety from Malaysia, two from Vietnam, 13 from Indonesia, 41 from Japan, 51 from China, two from Cambodia, one from Thailand, two from the Philippines, and 31 from Korea. Group III accounted for about 86% of the rice genetic resources of Japan, China, and Korea. In Group IV, Korean rice genotypes constituted the majority, with one variety from Myanmar, one from Indonesia, four from Japan, 10 from China, one from Cambodia, one from Thailand, one from the Philippines, and 35 from Korea. Group IV was also divided into two subgroups (IV-1, IV-2); Subgroup IV-1 had one variety from Indonesia, one from Japan, five from China, one from Cambodia, one from Thailand, and 31 from Korea. In Subgroup IV-2, there was one variety from Myanmar, three from Japan, five from China, one from the Philippines, and four from South Korea. Subgroup IV-2 contained rice genes from East Asian countries including Japan, China, and Korea. The UPGMA analysis classified most of the Southeast Asian rice genetic resources into Group I. Rice genetic resources of Japan were included in Group II, and Group III contained genetic resources from Japan, China, and Korea. In Group IV, most of the samples were confirmed as Korean rice genetic resources. These groupings were based on areas where each rice genetic resource was grown, mainly Southeast Asia and East Asia. Southeast Asia, because of its proximity to the equator, represents a tropical climate with high temperatures throughout the year, and it is influenced by the monsoon season with frequent precipitation, except in the dry season. Therefore, it was confirmed that the ssp. *indica* varieties, suitable for this environment, were cultivated in Southeast Asia. On the contrary, the East Asian region has four distinct seasons, spring, summer, autumn, and winter, and has many plains; in this climate, the ssp. *japonica* varieties are cultivated rather than ssp. *indica*. An analysis of the dendrogram of the microsatellite marker genotypes from this study confirmed that similar rice varieties were in the same groups, and similar varieties of crosses were used for the cultivation of growth environments and breeds. When the dendrogram was made by the UPGMA analysis, Group I contained genetic resources cultivated in Southeast Asian countries, such as Indonesia, Malaysia, the Philippines, and Thailand. Groups I, III, and IV included genes cultivated in the East Asian countries of Japan, China, and Korea. These results were divided according to the genetic resources of the maternal and paternal families from which the varieties were developed as well as the results of other researchers and the genotype of rice or the environment of the region where the cultivars were grown [28,31]. Therefore, it is expected that more effective breeding results will be obtained by selection based on the maternal or paternal samples when cultivating a new breed for a new environment.

### 2.3. Population Structure

A model-based program called STRUCTURE v2.3.4 was used to determine the genetic relationship of the 548 colored rice genetic resources. In order to analyze the colored rice genetic resources, ΔK was chosen with the peak alpha parameter. Population structure of the 192 germplasm lines was analyzed by a Bayesian based approach. The estimated membership fractions of 192 accessions for different values of k ranged between 2 and 5 (Figure 3). The log likelihood revealed by structure showed the optimum value as 2 (K = 2). However, when the K value was set to 2, 548 colored rice gene sources could not be effectively distinguished. Because 548 colored rice gene sources have a very diverse gene pool. Thus, when the K value was set to the next higher K = 3, 4, 7, the population structure was confirmed. When the K value was set to 7, 548 colored rice genes were classified into the optimal population structure. The seven groups were divided according to the environments in which the colored rice genetic resources were grown and the type of classification. In addition, analysis of the population structure distinguished whether the colored rice genetic resources belonging to the group were pure or admixtures; varieties with a probability value greater than 0.80 were considered pure. Most of the colored rice genetic resources belonging to cluster 7 were pure, and most of the colored rice genetic resources belonging to the remaining subgroups were admixtures (Figure 4). Most of the colored rice genetic resources used in this study were in the form of admixtures, meaning that the colored rice genetic resources were not high in purity but derived through many recombination and crossbreeding events. Population structure analysis showed that the grouping of colored rice genetic resources was like that of the dendrogram for the UPGMA analysis based on genetic distances. When the dendrogram was drawn through the UPGMA analysis, it was divided into four groups with seven subgroups. The population structure analysis showed the highest value at K = 7, so the 548 colored rice genetic resources were divided into seven groups. Population structure analysis showed that the classification of colored rice genetic resources, the geographical location, and the goal of breeding were significantly influenced by the genetic structure of rice genotypes.

### 2.4. Principal Component Analysis (PCA)

PCA is an analytical method that finds the best representations of the differences of each dataset and distinguishes the data by each element. In other words, PCA is a method in which when data are presented as axes, the axis with the greatest variance is set as the first main component, and the axis with the second largest variance is displayed as a diagram. When PCA was performed to confirm the degree of genetic diversity of the 548 colored rice genetic resources, all the genetic resources were uniformly distributed in four quadrants and were not shifted to any one place (Figure 5). Two-dimensional data obtained through PCA analysis resembled the results of the dendrogram and population structure analyses, and most varieties were distributed in all quadrants. When the UPGMA dendrogram was created, the colored rice genetic resources belonging to Subgroup I-1 were uniformly distributed in the first and second quadrants of the diagram, and those of Subgroup I-2 were distributed in the third quadrant. Subgroup I-3 appeared in the lower-left corner of the fourth quadrant, and Group II appeared in the upper-left quadrant. Group III was shown in the lower half of Quadrant 1 and Quadrant 4. Subgroup IV-1 was uniformly shown in the first and second quadrants like Group I, and Subgroup IV-2 was shifted to the lower right of the third quadrant. This PCA can be used to classify genotypes of colored rice genetic resources based on molecular morphological changes as well as plant morphological data. However, the PCA only finds the best representatives of the differences in genetic resources in the population; it does not provide information on the number of populations or subgroups of populations. In addition, PCA is based on morphological markers and it analyzes group structure. Therefore, PCA is less accurate than other methods. Since PCA showed very high variability in the principal component of this study, this information may be helpful in identifying related genotypes if used with population analysis. It can also be useful to distinguish between subgroups of populations involved in classifying genotypes. In addition, when comparing the PCA with the dendrogram, they showed similar tendencies and their general and group compositions were similar.

## 3. Discussion

Sixteen SSR markers were used for the assessment of colored rice genetic diversity. Each marker can distinguish only one or two varieties, but using multiple marker set simultaneously can assess the genetic diversity of numerous gene resources. In this study, when UPGMA dendrograms of the 548 colored rice genetic resources were created using Jaccard’s distance coefficients and the MEGA7 program, they were divided into four large groups (I, II, III, and IV) and seven subgroups (I-1, I-2, I-3, II, III, IV-1, and IV-2). Moreover, when analyzing the population structure using the STRUCTURE v2.3.4 program, it was confirmed that as in the previous results, optimal separation occurred when the 548 colored rice genetic resources were divided into seven subgroups [32]. In the analysis of the colored rice population structure using the UPGMA dendrogram and STRUCTURE programs, it was confirmed that it was divided into seven groups in all analysis. The results were obtained when assessment genetic diversity based on genetic distance and model-based population structure. In addition, when analyzing the population structure using the STRUCTURE program, it was confirmed that the colored rice gene sources were genetically mixed. The reason why the genetic material is so mixed is because breeding continues to develop varieties with high yield and quality. Since this breeding will not end now and will continue in the future, the mix of genetic resources will continue to increase. When the UPGMA dendrogram was created, Group I contained evenly colored rice genetic resources from Southeast Asian countries such as the Philippines, Indonesia, Thailand, Korea, China, and Japan. Group II included most of the Japanese colored rice genetic resources, and Group III included the rice genetic resources of East Asian countries including Japan, China, and Korea. Finally, Group IV included most of the Korean colored rice genetic resources. When the selected microsatellite markers were used to analyze the populations of rice genetic resources, some of the varieties were classified as identical. It was presumed that these results were obtained because rice genetic resources are composed of the same maternal or paternal material and the genetic resources should be judged by either comparing the morphological characteristics or increasing the number of microsatellite markers. In this study, 16 microsatellite markers were newly selected from the 600 pairs of microsatellite primers reported in the rice genome database (http://www.gramene.org/); they showed high polymorphism and excellent repeatability of the pattern of alleles from the PCR results. The results of the analysis of the 548 colored rice genetic resources, the black-purple rice and red pericarp rice, using these microsatellite markers showed that the colors were not clearly distinguishable but that the resources were clearly grouped according to plant taxonomic characteristics, breeding lineage, and cultivation environment of rice. Therefore, the database of DNA profiles constructed through the selected microsatellite markers can be used not only to evaluate the characteristics of rice genetic resources but also to select the controls for cultivar-protected varieties. There are many genetic resources that have not been identified by the microsatellite markers used in this study and there have been many recent reports of new SNP markers [33], highlighting the necessity to continue to develop new markers, specific to varieties, by comparing and analyzing morphological characteristics.

## 4. Materials and Methods

### 4.1. Plant Material

In this study, 548 colored rice genetic resources (376 black-purple rice samples, 172 red pericarp rice samples) collected from national and international sources were used as analytical materials for microsatellite markers (Appendix A).

### 4.2. Genomic DNA Extraction

Genomic DNA was isolated using a NucleoSpin^®^Plant II (Cat. 740 770.250, Macherey–Nagel GmbH & Co., KG, Deutsch, Düren, Germany) kit. The extracted genomic DNA was quantitated at 260 nm using a spectrophotometer, and the DNA concentration was confirmed by electrophoresis on 1.5% agarose gel. The DNA was diluted to a final concentration of 20 ng/μL using sterilized water and then used for SSR analysis. The concentration and quality of the obtained genomic DNA samples were estimated with an ultramicrospectrophotometer Nano Drop (Cat. ND-2000, ThermoFisher SCIENTIFIC, Seoul, South Korea).

### 4.3. SSR Genotyping

In order to select SSR markers effective for colored rice genetic resources, the 600 SSR primer sets reported in the rice genome database (http://www.gramene.org/) were used to carry out PCR on the “Hopum,” “Hwayeong,” “Dami,” “Sindongjin,” “Odae,” “Chucheong,” “Saechucheong,” “Junam,” “Ilpum,” “Ilmi,” and “Gopum” rice varieties. The primers with high repeatability, high band reproducibility, and a high degree of polymorphism were first selected. Primers having a high PIC value and a high heterozygosity value were selected among the primers selected first. Finally, the selected SSR primer sets were distributed evenly on the rice chromosome. Gene amplification products by PCR were analyzed by electrophoresis using Genetic Analyzer (HAD-GT12TM System, *e*GEnE, Irvine, CA, USA), and the differences in the various alleles were analyzed using a computer program (Biocalculator 2.0). The PCR amplification results were used to construct a DNA profile database for the 548 colored rice genetic resources. The PCR reaction mixture contained 50 μg of genomic DNA, 0.1 μM of SSR primer, and 2.0 μL of dNTP mixture (2.5 mM), and 1.0 U of HS Prime Taq polymerase (Cat. G-7002, Genet Bio, Daejeon, South Korea, 50 mM KCl, 20 mM Tris-HCl, pH 8.0, 2.0 mM MgCl 2) was added with distilled water to adjust the total volume to 30 μL. PCR amplification (C1000, BioRad, Hercules, CA, USA) was performed by denaturation at 94 °C for 30 s, annealing at 55 °C for 30 s, and extension at 72 °C for 45 s, for a total of 40 cycles. After completion of the PCR, 3.0 μL of the PCR amplification product was electrophoresed on a 2.5% agarose gel to confirm the PCR amplification. After confirming amplification, 1.5 μL of the PCR amplification product was diluted in 150 μL of distilled water. To estimate the size of the alleles of the chromogenic genes according to the SSR markers, 1.0 μL of the PCR amplification product, 10 μL of deionized formamide, and 0.25 μL of size marker (LIZ500 size standard) were mixed. After denaturation at 94 °C for 2 min, electrophoresis was performed using an automatic sequencer (Genetic Analyzer 3130XL, Applied Biosystems, Foster City, CA, USA). And the size of amplified alleles was measured using the Gene Mapper (version 3.7) program (Applied Biosystems, Foster City, CA, USA). The number of alleles and the size of alleles were determined for each marker.

### 4.4. Genetic Diversity Analysis

PIC values were calculated using the following formula to investigate the diversity of microsatellite markers:(1)$$\mathrm{PIC}=1-\overset{n}{\underset{i=1}{\sum }}p_{i}^{2},$$
where *n* is the number of alleles analyzed per marker and Pij is the frequency of the jth common band pattern among the bands of marker *i*. [34]. Microsatellite analysis was used to select alleles with high reproducibility and high polymorphism as markers. NTSYSpc (version 2.10) [35] was performed according to dominant marker scoring (present = 1, absent = 0), and genetic similarity values were calculated according to the Jaccard’s method [36], followed by population analysis using UPGMA [37]. The population structure was deduced from the model-based program STRUCTURE v2.3.4 [38] and performed over the range of K = 2 to 10. The final K value was determined using Evanno’s ΔK method [39]. Allelic frequencies of each microsatellite marker and each variety were calculated and used for PCA. PCA analysis was performed using the R program (ver. 3.2.3).

## Figures and Tables

**Figure 1 plants-08-00471-f001:**
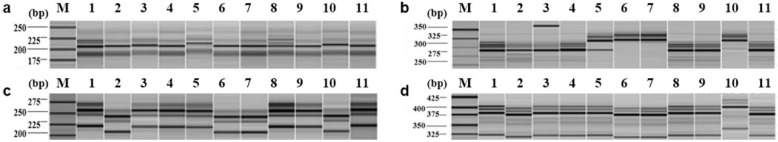
Polymorphism of four simple sequence repeat (SSR) markers, RM20754, HVSSR2-86, RM15641, and RM19731. The amplified polymerase chain reaction (PCR) products were loaded on the HAD-GT12™ Genetic Analyzer System and analyzed using Biocalculator Data Analysis Software. (**a**): RM20754, (**b**): HVSSR2-86, (**c**): RM15641, (**d**): RM19731; Lane 1: Hopum, 2: Hwayeong, 3: Dami, 4: Sindongjin, 5: Odae, 6: Chucheong, 7: Saechucheong, 8: Junam, 9: Ilpum, 10: Ilmi, 11: Gopum.

**Figure 2 plants-08-00471-f002:**
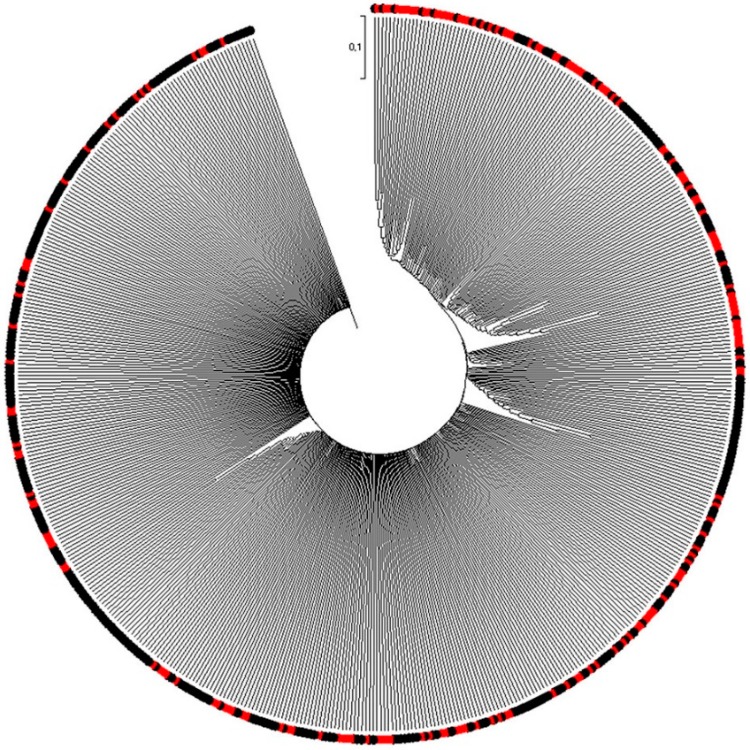
Unweighted pair-group method with arithmetical average (UPGMA) dendrogram of 548 rice germplasms based on 16 SSR markers using MEGA7 program. The red color in the boundary indicates rice genotypes that have red seeds and the black color indicates rice genotypes that have black seeds. Black and red rice are not clearly distinguished and mixed in large groups.

**Figure 3 plants-08-00471-f003:**
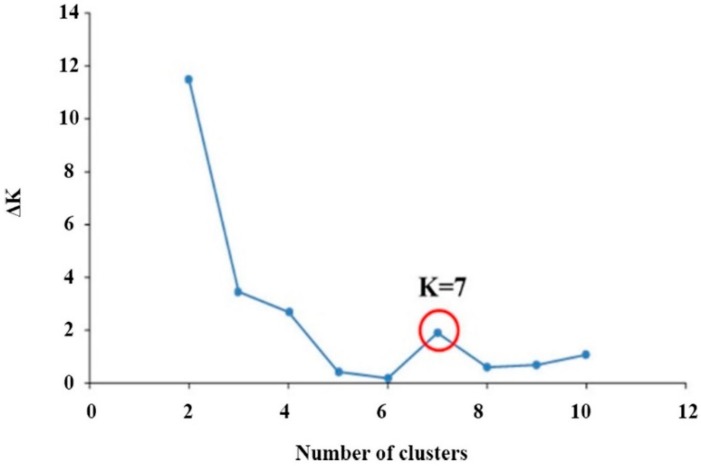
Inference of number of clusters in rice germplasm collection based on 16 SSR markers using STRUCTURE program. The Bayesian model in the STRUCTURE v2.3.4 program was used to test K (number of clusters) using the parameter of burn-in/Markov chain Monte Carlo (MCMC): 100,000/250,000 iterations. An ad hoc quantity (ΔK) [31] was calculated for each K to detect the best number of clusters (K = 7).

**Figure 4 plants-08-00471-f004:**
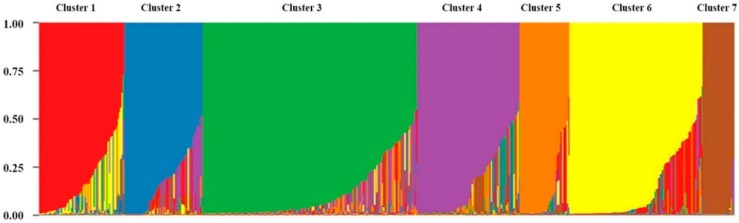
Structure plot presenting seven clusters of rice germplasm collection. The *y*-axis indicates the estimated membership coefficients for each individual. Each variety’s genome is represented by a single vertical line, which is partitioned into colored segments in proportion to the estimated membership in the seven clusters.

**Figure 5 plants-08-00471-f005:**
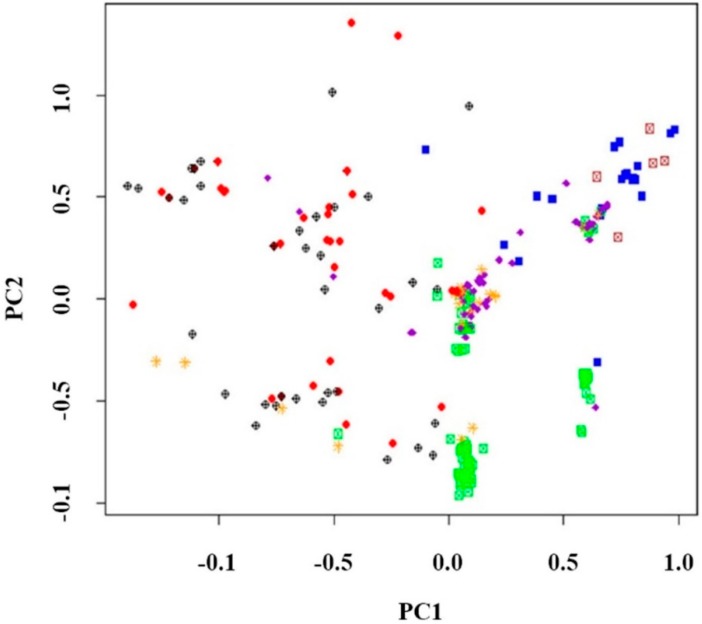
Principal component analysis (PCA) of 548 rice germplasms based on 16 SSR markers using R program. The colors shown in this figure represent the same genetic resources as the colors in Figure 5. The rice genotypes belonging to cluster 1 are indicated by red; cluster 2: blue; cluster 3: green; cluster 4: violet; cluster 5: orange; cluster 6: black; and cluster 7: brown.

**Table 1 plants-08-00471-t001:** Repeat motif, no. of alleles, and polymorphism information content (PIC) value of microsatellite markers selected for genetic characterization of rice cultivars and germplasms.

No.	SSR Primers	Repeat Motif	Chromosome Number	Annealing Temperature (°C)	PCR Product Size (bp)	No. of Alleles	PIC Value
1	KRF24	terminal repeat	7	55	213–238	22	0.902
2	RM8085	ag	1	55	105–144	20	0.915
3	RM72	(tat)_5_c(att)_15_	8	55	151–207	20	0.916
4	KRF16	aac	9	55	160–253	27	0.963
5	HvSSR02-86	aac	2	55	187–328	47	0.970
6	RM18821	tct	5	55	150–205	17	0.876
7	RM12834	aga	2	55	186–260	25	0.862
8	RM20754	aat	6	55	151–219	22	0.955
9	RM26063	ct	11	55	225–283	27	0.952
10	RM333	aat	10	55	156–312	35	0.964
11	RM15641	aat	3	55	163–247	31	0.894
12	RM26730	ctt	11	55	98–157	11	0.781
13	RM19731	tta	6	55	339–429	31	0.964
14	KRF15	atg	8	55	126–175	18	0.878
15	KRF17	tatc	2	55	243–405	41	0.956
16	RM24946	atg	10	55	317–369	15	0.855
Total						409.00	14.604
Mean						25.56	0.913

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
