# Peer review of "Assessment of the Genetic Diversity of Rice Germplasms Characterized by Black-Purple and Red Pericarp Color Using Simple Sequence Repeat Markers"

_plants, 2019, doi:10.3390/plants8110471_

Round 1
Reviewer 1 Report
Requested corrections were carried out by authors.
Author Response
Thank you for your kind comment. Thank you, sincerely.
Reviewer 2 Report
Hi author,
I appreciate the changes made in the last round of revision. However, most of the figures, though they are colorful, are still not informative to the audience.
Table 1 is fine now.
Figure 1 shows some gel images but as a reader/reviewer, I don't know where to look for because it is still difficult to interpret the gels when they show everything from 0 bp to 475 bp. Please focus on only regions of interest.
Figure 2 still has the same problem as pointed out in the previous round. As a reader/reviewer, all I see is a complicated dendrogram but am unable to make sense of it. I propose that you zoom in on certain regions as a main figure here, and then show the entirety in supplementary section.
Figure 3 has the same concern, where the figure is visually appealing but lack information for the audience. What is the main point and does the figure successfully convey that? If some clades need to be annotated, please do. Otherwise, this figure should be discarded.
The main concern I have with Figure 4 is not knowing how you decide that "K =7" is the best number of clusters. In your revised manuscript (line 223 to 225), you mentioned the comparison of LnP (D) values, which do not correspond to the y-axis of Figure 4.
Figure 5 is also another aesthetically pleasing figure that is need of proper annotation. Here is a suggestion: can you include 7 groups of rice varieties that correspond to this colored figure? A reader cannot make sense of the figure, as it is.
Figure 6 is PCA plot lacking legend description. I see a green cluster and a moderate spread-out blue cluster. But what do these colors correspond to? Please include rice genotypes/varieties/accessions that correspond to the colored clusters of the PCA plot. Additionally, have you explored other PC-biplot (PC1 vs PC3, PC2 vs PC3)?
Author Response

(The authors gave the same response as above.)

Round 2
Reviewer 2 Report
Hi author,
Thanks for addressing my concerns. The manuscript and content look fine now.
This manuscript is a resubmission of an earlier submission. The following is a list of the peer review reports and author responses from that submission.
Round 1
Reviewer 1 Report
The evaluating of the genetic diversity within germplasm collections can be accomplished using simple sequence repeat (SSR) markers and association mapping techniques. The results provided valuable information for the colored rice breeding program and showed the importance of protecting germplasm resources and the molecular markers that can be derived from them. Though the overall interest and visibility of this work, some aspects should still be considered to improve the quality and objectiveness of this work.
Background of the study should be made to very clear. Provide more details of introduction and review of the work. Materials and methods section provide detail information. Overall, this manuscript needs more discussion about experimental results. Please speculate about the reasons to the obtained results. Interpretation is not enough. Need to discuss in details. Need to provide recent references. In Conclusion, authors should add significance of this research to potential practical application. Overall, this manuscript written is very poor.Author Response
Response to Reviewer 1 Comments
Point 1: Background of the study should be made to very clear. Provide more details of introduction and review of the work.
Response 1: We have added a description of the necessity of this study in the Introduction. Added content is shown in blue.
Point 2: Materials and methods section provide detail information.
Response 2: Added content to DNA extraction and SSR genotyping of materials and methods. The added part is marked in blue.
Point 3: Overall, this manuscript needs more discussion about experimental results. Please speculate about the reasons to the obtained results.
Response 3: Added content to discussion. The content I added has been marked in blue.
Point 4: Interpretation is not enough. Need to discuss in details.
Response 4: Corrected the wrong part through the correction. Attach proof of calibration documents.
Point 5: Need to provide recent references.
Response 5: Updated references 3, 4 and 21 to recent references. The modified reference is shown in blue.

Reviewer 2 Report
The English needs tightening up in general, please think about having it edited professionally.
Be careful about alleles versus loci. For example line 31 should read 16 loci or 409 alleles, not 409 loci
Line 61 - how has domestication caused a reduction in diversity in wild rice?
Line 69 - can you add a citation about these disputes?
Line 110 - how were they "identified from 11 different rice"? - I think you mean they were amplified not identified. And as the methods come at the end of this paper it would be better to explain this here, for example "The 600 loci were amplified from eleven accessions..."
Lines 124-6 - Its unusual to score these factors 'per chromosome', why weren't they calculated per marker?
You say these markers have high PIC and high variation, but in the image, marker RM20754 appears to be monomorphic. Is this correct? It also appears 15641 only has two alleles, again suggesting very little variation. I think you need to annotate these figures to show exactly what you are scoring ( see final point below about the methods). And zoom in further so allelic variation can be seen. At the moment only large differences would be apparent because the Y axis goes from 25-400. Show only the relevant part, eg 150-200 for RM20754.
Table 1 - why is some data missing? Surely you can find the repeat motif and the chr number? Row 5 is missing the '5'
Figure 2 is useless - the reader cannot read it and there are nine panels without annotation. Make this a big supplement if you can't fit it onto one page like a properly formatted tree in a manuscript.
Lines 158-183 - please put this in a table so it is easy to understand. Make each row one of your subpopulations (I-1, I-2... IV-2) and each column a geographic region or a country. The in each cell put the number of accessions from that country found in that subgroup. You should also do a statistical analysis to show whether this is random or not. You can do a chi2 test to show if the accessions are geographically random with respect to their genetic cluster, or if there is deviation, and hence clustering by geography.
Figure 3 is not much better than figure 2 - what is a reader supposed to be able to see here? At least annotate it with the clusters you've described!
L213 - please rewrite this, its confusing. What does "values of LnP(D) and Evanno" mean? Evanno is a person.
L214 - incorrect. It is highest at K=2, second highest at 3, third highest at 4 and only fourth highest at 7! Picking 7 is not good here. At least show the results for 2, 3, 4, 7 as this means something in a hierarchical fashion (read the paper in Mol Ecol by Meirmans: https://onlinelibrary.wiley.com/doi/full/10.1111/mec.13243). Edit also lines 284-5
Figure 5 - what is the reader learning from this? Without knowing which accessions are in which of the 7 clusters then there is no info here of use.
The structure results can also be used to determine how many accessions are 'pure' versus 'admixed'. please add this information, it is interesting.
Lines 249-252 can be deleted.
How did the structure results compare to the UPGMA clusters? Did they overlap?
Section 2 should not be "results and discussion" if section 3 is "discussion". There is a lot of discussion in the results that should be moved to the discussion. For example the bottom of page 5 needs moving.
You mention "repeatability" several times but I don't see anywhere that you tested this. Please add this.
Line 351 - A good SSR marker should show one or two alleles in a diploid rice individual. Did you score every band from each marker or just one or two from a specified locus? The latter is a more reliable method because it is know that a single locus is being investigated, not many loci. Please clarify and if necessary think about reanalysing.
Author Response
Response to Reviewer 2 Comments
Point 1: Line 61 - how has domestication caused a reduction in diversity in wild rice?
Response 1: Thank you for your kind comment. People selected and cultivated high-quality, high-yield rice. Thus, poor quality and low yield varieties were eliminated during this process. Therefore, the genetic diversity of the rice was reduced.
Point 2: Line 69 - can you add a citation about these disputes?
Response 2: Thank you for your kind comment. We have added a reference to the paper containing SSR markers to assess gene diversity and help resolve seed disputes. Added as the 13th reference, the modified part is marked in red.
Point 3: Line 110 - how were they "identified from 11 different rice"? - I think you mean they were amplified not identified. And as the methods come at the end of this paper it would be better to explain this here, for example "The 600 loci were amplified from eleven accessions..."
Response 3: Thank you for your kind comment. Many SSR markers have already been developed. We sought to select the optimal marker to distinguish between 548 colored rice gene sources. We applied 600 SSR markers that were already developed in 11 representative varieties of Korea (Hopum, Hwayeong, Dami, Sindongjin, Odae, Chucheong, Saechucheong, Junam, Ilpum, Ilmi, Gopum”). Applying 600 SSR markers to 548 colored rice gene sources is costly and time consuming. Therefore, 600 SSR markers were applied to 11 representative varieties, and 16 SSR markers with the highest polymorphism and efficiency were selected. Among them, 16 SSR markers, which are highly diverse and dominant, were selected.
Point 4: Lines 124-6 - Its unusual to score these factors 'per chromosome', why weren't they calculated per marker? You say these markers have high PIC and high variation, but in the image, marker RM20754 appears to be monomorphic. Is this correct? It also appears 15641 only has two alleles, again suggesting very little variation. I think you need to annotate these figures to show exactly what you are scoring (see final point below about the methods). And zoom in further so allelic variation can be seen. At the moment only large differences would be apparent because the Y axis goes from 25-400. Show only the relevant part, eg 150-200 for RM20754.
Response 4: Thank you for your kind comment. Not only one SSR marker was used to assess the diversity of rice genetic resources. If we used RM20754 or RM15641 markers to assess the diversity of genetic resources, we would not be able to identify them. But we used various of SSR markers to evaluate the diversity of the colored rice gene resource. And using these data collectively to assess the diversity of the genes, we were able to classify all the genetic resources according to their genetic distance, and we could create a dendrogram like Figures 2 and 3.
Point 5: Table 1 - why is some data missing? Surely you can find the repeat motif and the chr number? Row 5 is missing the '5'
Response 5: Thank you for your kind comment. We used several SSR markers already developed at the Rural Development Administration. And these markers are very useful for distinguishing the genetic diversity of rice. The reason we can see that it is a suitable SSR marker for evaluating genetic diversity is that we get high values by calculating PIC values. Some of these markers, however, do not yet have some information, such as repeat sequences and gene positions. Only information is known to help identify genetic diversity. Also, some SSR markers are very complicated in repeating positions. And No. of the table. Added 5 missing parts and marked in red.
Point 6: Figure 2 is useless - the reader cannot read it and there are nine panels without annotation. Make this a big supplement if you can't fit it onto one page like a properly formatted tree in a manuscript.
Response 6: Thank you for your kind comment. Figure 2 shows the dendrogram of 548 colored rice gene sources. This is the result of using 16 SSR markers. This data shows that the colored rice genetic resources are very diverse. You can also see that the colored rice genetic resources are divided into a wide variety of groups. However, the picture is so large that I can't see any details. So we've created a supplement and attached it. The word supplement figure has been added to line 154 and the revised portion is shown in red.
Point 7: Lines 158-183 - please put this in a table so it is easy to understand. Make each row one of your subpopulations (I-1, I-2... IV-2) and each column a geographic region or a country. The in each cell put the number of accessions from that country found in that subgroup. You should also do a statistical analysis to show whether this is random or not. You can do a chi2 test to show if the accessions are geographically random with respect to their genetic cluster, or if there is deviation, and hence clustering by geography.of mentioning gene expression level, the authors have mentioned the technique name.
Response 7: Thank you for your kind comment. There is already information about genetic resources in supplement table 1. Supplement Table 1 contain the countries and groups to which each genetic resource belongs. This is also explained in the results and discussion. The cluster portion of supplement table 1 is further subdivided and the modified portion is marked in red.
Point 8: Figure 3 is not much better than figure 2 - what is a reader supposed to be able to see here? At least annotate it with the clusters you've described!
Response 8: Thank you for your kind comment. Figure 3 is similar to Figure 2. Figure 3, a dendrogram was created using genetic distances. This is represented by a circle, and the border of the circle is marked in black and red to distinguish red and black rice. Therefore, because the dendrogram was drawn using genetic distance, the black and red rice were not clearly distinguished. Thus, black rice and red rice indicate that there is not much genetic difference. The additions to the legend in Figure 3 are shown in red.
Point 9: L213 - please rewrite this, its confusing. What does "values of LnP(D) and Evanno" mean? Evanno is a person.
Response 9: Thank you for your kind comment. I modified the overlapping meaning of the sentence. The modified part has been marked in red.
Point 10: L214 - incorrect. It is highest at K=2, second highest at 3, third highest at 4 and only fourth highest at 7! Picking 7 is not good here. At least show the results for 2, 3, 4, 7 as this means something in a hierarchical fashion (read the paper in Mol Ecol by Meirmans: https://onlinelibrary.wiley.com/doi/full/10.1111/mec.13243). Edit also lines 284-5
Response 10: Thank you for your kind comment. When k is 7, it does not represent the highest value. Only when k is 7, peaks appear, which means that the best separation occurs when k is set to 7. These results indicate that when the dendrogram was drawn using genetic distance, 548 colored rice genetic resources were divided into seven groups (I-1, I-2, I-3, II, III, IV-1, IV-2). The same result.
Point 11: Figure 5 - what is the reader learning from this? Without knowing which accessions are in which of the 7 clusters then there is no info here of use. The structure results can also be used to determine how many accessions are 'pure' versus 'admixed'. please add this information, it is interesting.
Response 11: Thank you for your kind comment. In this graph, the x-axis represents genetic resources and the y-axis represents probability values. And if one color is over 50%, this gene source is pure, otherwise it is admixture. There are also pure genetic resources, but a lot of admixture genetic resources were distributed. The reason for the large number of admixtures is because the continuous breeding develops varieties with higher yields and higher quality than existing gene sources.
Point 12: Lines 249-252 can be deleted.
How did the structure results compare to the UPGMA clusters? Did they overlap?
Section 2 should not be "results and discussion" if section 3 is "discussion". There is a lot of discussion in the results that should be moved to the discussion. For example the bottom of page 5 needs moving.
Response 12: Thank you for your kind comment. The structure analysis also showed the same results as the previous results. In all analyzes, 548 colored rice gene sources were divided into seven groups.
Point 13: Line 351 - A good SSR marker should show one or two alleles in a diploid rice individual. Did you score every band from each marker or just one or two from a specified locus? The latter is a more reliable method because it is know that a single locus is being investigated, not many loci. Please clarify and if necessary think about reanalysing.
Response 13: Thank you for your kind comment. The markers used in this study were used to detect only one or two fragments in a specific region. The reason for this is the convenience of the analysis.

Reviewer 3 Report
Hi author,
Thanks for the well-conceptualized and written manuscript. It was overall a pleasure reviewing your manuscript. My comments are mostly minor, but they will help in improving your manuscript.
See line 235. It is not clear how "inference of number of clusters" was obtained. This is only briefly explained in "Materials and Methods" section (lines 363-265). Some descriptions on what the "K value" tells us would be helpful. Legibility of figures.Figure 1 is okay (marker bands not legible though).
Figure 2 needs a major overhaul; the numbering on the right and the bottom scale cannot be read.
Figure 3 is okay (should increase the size of "0.1").
Figure 4 needs some improvements; figure is pixelated. Please use vector-based image instead of rasterized (if unsure, look up "raster vs vector").
Figure 5 also needs to be re-generated; figure is pixelated and appears to be shrunk vertically (look at the y-axis labeling).
Figure 6 can be improved. Texts appear smeared.
Author Response
Hi,
Please see the attachment.
Thank you for review my paper.

Round 2
Reviewer 1 Report
Requested corrections were carried out.
Author Response
Hi,
Point 1: Requested corrections were carried out.
Response 1: Thank you for your kind comment. Your review was very helpful. Thank you.
Thank you for review my paper.
Reviewer 2 Report
The authors have only made small changes and have ignored several comments.
The figures are still poor and useless, even the extra figure lacks detail for the reader to interpret it.
The interpretation is still lacking (presenting more than one value of K for structure is important and has been ignored, similarly, the lack of thorough comparison between the different analyses is still absent). I am sending it back to the authors hoping they will actually address my comments this time.
Here is my full response to the response:
Reviewer 2 Point 1: Line 61
I know how domestication works. Cultivated rice has less diversity than wild rice, but your statement says there is now less diversity in WILD rice. How is there now less diversity in wild rice? What you seem to mean is that during domestication the genetic diversity becomes reduced such that cultivated rice has less diversity than wild rice. But this is not what the statement says.
Point 2: Line 69 - you haven't addressed my comment. In the response to reviews you have written about something else. My point is about individual markers having low polymorphism despite you saying that markers with high variability were selected.
Reviewer 2 Point 3: Line 110
The SSR markers were amplified in 11 varieties, they weren't identified from 11 varieties.
Reviewer 2 Point 4: Lines 124-6
This comment was ignored by the authors. Please address both aspects of this ('per chromosome is wrong' and the figure shows very little diversity in your 'very diverse' markers)
Reviewer 2 Point 5:
Why cant you find this locus in the genome to see what repeat is present? Where is this information from and why doesn't it say where in the genome the locus is? This is important - what if two loci were next to each other in the genome?
Reviewer 2 Point 6:
The authors have made no attempt to improve this figure and so the in-text figure is still useless. There isnt even an attempt to explain which panel follows on from which If the authors cant be bothered to improve it then just remove it.
Adding the supplement is useful, but it is poorly drawn. Please remove the scale bar from each panel and line up the panels. In addition indicate which accession is which (don't just give the number) and add colour to indicate the red and black varieties. Also you mention how many groups there are but you have made no attempt to indicate these in the figure. Please add this too.
Reviewer 2 Point 8:
Again, the authors have not improved this. Why show both figures 2 and 3? they have the same information, right? And figure 3 is unclear too. Adding a sentence to the legend does not make the lines clearer to see.
Reviewer 2 Point 10
My original comment stands. In an LnP(D) plot one is looking for the highest LnP(D) (which you even say in your text), which occurs at 2, then 3, then 4, with K=7 being the fourth highest, and therefore K=7 is not the optimal K.
Also read the paper I sent don't ignore it. One should not only present the results for one value of K.
Reviewer 2 Point 11
I don't need you to tell me how to interpret a structure plot. I have drawn a hundred of these. My comment refers to the figure being useless without proper annotation. Please read my comments properly and act on them, e.g. "Without knowing which accessions are in which of the 7 clusters then there is no info here of use."
Reviewer 2 Point 12:
First, structure DOES NOT show K=7 is optimal. Second I can read - I know you say both results have 7 groups. But are the same accession in the same groups in both analyses. I also want to see this written in the text, not in a comment to the reviewers only.
Line 295 - " confirming that the accuracy of this study is high" - I don't agree. Accuracy is about doing it right, not about getting the same result twice. You could have done it all incorrectly and found the same result twice. What you mean is there is consistency. HOWEVER without trying to answer my previous comment (Reviewer 2 Point 12) there is no evidence for this.
Line 333 - were the five individuals pooled or analysed separately? Please be precise in your descriptions.
Author Response

(The authors gave the same response as above.)

Round 3
Reviewer 2 Report
The authors have still not addressed my comments and so I am not going to waste my time saying them for a third time. I have listed things that need changing that have not been changed. Its up to the editor now to pursue these changes (highly recommended) or not.